# Research on the Single-Value Indicators for Centrifugal Pump Based on Vibration Signals

**DOI:** 10.3390/s20113283

**Published:** 2020-06-09

**Authors:** Yin Luo, Yuejiang Han, Shouqi Yuan, Jianping Yuan

**Affiliations:** Research Center of Fluid Machinery Engineering and Technology, Jiangsu University, Zhenjiang 212013, China; 2221711011@stmail.ujs.edu.cn (Y.H.); shouqiy@ujs.edu.cn (S.Y.); yh@ujs.edu.cn (J.Y.)

**Keywords:** centrifugal pump, single-value indicators, operation condition monitoring, vibration, hydraulic instability

## Abstract

Off-design operation conditions might not only seriously affect the internal flow status of a centrifugal pump, but also result in additional energy loss and potential mechanical damage. Therefore, early-stage monitoring and predication on off-design operation conditions for centrifugal pumps have become essential. Single-value indicators have favorable factors such as a smaller amount of calculation and easier identification. As a result, industries prefer the more straightforward approach: obtaining single-value indicators directly from the signals which could be easier compared with accepted standards. The possibility of applying the single-value indicators of vibration into operation condition monitoring for a centrifugal pump is studied theoretically and experimentally, which shows that the statistical features of vibration might be suitable for hydraulic instability detection for a centrifugal pump.

## 1. Introduction

A centrifugal pump is supposed to keep operating under its design operation point, or best efficiency point (BEP), which is the specific combination of head, flowrate and rotational speed. Under the BEP, fluid motion of a centrifugal pump and physical contours of hydraulic passages would basically coincide in the main. Nevertheless, quite a number of centrifugal pumps usually deviate from their optimal operation points in industries [1]. Smaller flowrate than the design flowrate means the centrifugal pump operates under part-load condition, which would result in hydraulic instability and other negative factors that might damage the pump given the increasing large blade inlet angle and channel cross section [2]. On the contrary, a centrifugal pump operates under a larger flowrate than its design flowrate would increase the risk of cavitation, which might lead to stronger vibration and noise, considerable energy waste, causing even further damage and fracture of blades. In general, a centrifugal pump that fails to operate under the BEP would make itself subject to increasing wear, thus shortening its operation life. Effective fault diagnosis is thus essential to lower the risk of operation and maintain the health of a centrifugal pump [3,4].

Traditional operation monitoring technologies for centrifugal pumps often require the use of many invasive sensors. Along with them are the problems of hard installation, complex cabling, high equipment costs and difficult maintenance. Given consideration of these limitations, sensors are seldom employed in reality. Therefore, a system capable of accurate detection of operation conditions and increasing the efficiency of fault diagnosis [5] for centrifugal pumps should be developed [6]. The basic mechanism of such system is to analyze and subsequently identify the vibration signal obtained from one single acceleration sensor, which gives the system the superiority of easy achievement with no need to change the plant structure. Therefore, a method that could extract highly useful operation information of a centrifugal pump from simple monitoring equipment is worth developing [7,8,9].

Plenty of investigation and discussion have been undertaken on the development of operation monitoring and fault diagnosis methods for centrifugal pumps based on the vibration signal. Most of the proposed monitoring methods realize their function by either extracting the fault feature frequency components [10], or calculating the energy variation of some specific frequency bands. Hamomd et al. [11] established a vibration based fault diagnosis method for centrifugal pump based on modulation signal bispectrum (MSB) analysis. The method performs good diagnosis ability in the impeller and bearing faults for a centrifugal pump. Both fault types and fault degree could be indicated through MSB analysis due to its special advantages of noise reduction and non-linearity demodulation. Sun et al. [12] employed a cyclic spectral analysis method to process the non-stationary vibration signals acquired from a centrifugal pump operated under fault conditions. Investigation results suggest that the cyclic spectral analysis could reduce the disturbance effects from noise and modulated signals. Seal damage characterization and cavitation characterization could be effectively extracted as diagnosis indicators based on such a method. Li et al. [13] firstly developed the multivariate multiscale symbolic dynamic entropy (MvMSDE) to track the fault characteristics of a centrifugal pump from the measured synchronous multi-channel vibration signals, whose results could be selected as the input parameters of logistic regression (LR) for identification of fault types. Siano et al. [14] used the artificial neural networks (ANN)-based non-linear autoregressive (NLAR) approach to predict pump system behavior, and further offer the online detection on any abnormal behavior changes caused by cavitation operations. The proposed method had high detection performance of incipient faults during pump operation. Xue et al. [15] managed to build an intelligent diagnosis method for a centrifugal pump system. A support vector machine (SVM), possibility theory, and Dempster–Shafer theory (DST) were employed to process the vibration signals. Practical examples indicate that the method could effectively diagnosis faults and recognize fault types for centrifugal pump systems. Sakthivel et al. [16] established a fault diagnosis approach for centrifugal pump based on soft computing techniques. Specifically, the gene expression programming (GEP), wavelet-GEP, support vector machine (SVM) and proximal support vector machine (PSVM) were used for fault identification. The proposed diagnosis approach could realize the automatic online operation monitoring, which would greatly reduce the overall down time of centrifugal pump. Mohanty et al. [17] combined vibration analysis with motor current signature analysis for fault detection of a centrifugal pump. In their research, height of the sidebands and the difference between the amplitudes of vane pass frequency and sidebands in the vibration spectrum were used as the indicators to evaluate the fault severity for a centrifugal pump. Sakthivel [18] used C4.5 decision tree algorithm to extract and classify the statistical features from vibration signals of a monoblock centrifugal pump. The algorithm was verified to have high classification accuracy of pump faults.

However, industries are more inclined to the simple approach, which should have the advantages of less calculation and easier identification. Single-value indicators could be directly obtained from signal, their calculation procedures are relatively simple, their change rules could reflect more clearly the operation tendency of equipment, and their values could be easily compared against accepted standards. From this point of view, single-value measurements could be more suitable for deployment in industries than formal methods.

The most widely used single-value indicators of the vibration are the root mean square (*RMS*) and the peak value. They are often used as standards for vibration levels definition. As these indicators could show the energy intensity characteristics, when some adverse operating conditions occur, the energy intensity fluctuation might be indicated by them. The related application could be found in cavitation detection. However, sometimes the energy fluctuation features caused by the unsteady flow regime such as recirculation, secondary flow and even cavitation onset are too weak to observe. The existing single-value indicators would be insensitive to detect these adverse flow conditions under such circumstance. In this perspective, single-value indicators still lack the ability of extracting useful information.

Great efforts have been made to study the main characteristics of the vibration signal for centrifugal pumps. These studies indicate that the vibration signals from a centrifugal pump experiencing unsteady flow phenomena like cavitation are in a broad band with high frequency and a few discrete frequency components. The upper limit of the signal spectrum is more decided by the upper frequency limit of transducers than sources [19,20]. The specific spectral structure would be different at different flow regimes. Therefore, single-value indicators should have the ability to indicate the variation of the spectral structure.

In sum, this paper focuses on extracting the statistic characteristics of vibration signals from a centrifugal pump. Specifically, this study employs some statistical parameters as single-value indicators in order to reflect the pump’s operation status.

## 2. Theory and Methodology

### 2.1. Vibration Features of Centrifugal Pump

A centrifugal pump consists of two main parts: the rotating part and the stationary part; the former includes the shaft and the impeller, the latter includes the bearing, the casing, the casing box and the electrical motor with a cooling fan. Based on the operation mechanism of a centrifugal pump, vibration could be regarded as caused by both hydrodynamic and mechanical sources. Hydrodynamic sources usually lead to fluid flow disturbance, and facilitate the rotor–stator interaction, such as the interference between rotor blades and volute tongue or guide vanes. In contrast, mechanical sources are a result of shaft misalignment, unbalanced rotating mass and continuous friction in bearing and seals. Both kinds of sources would force the pump structure to vibrate. In this sense, the underlying generating mechanisms for both structure-borne vibration and airborne noise could be seen as the same for a sealed pump system [21,22,23].

Previous research has shown that that the vibration from a centrifugal pump contains a broadband noise and several discrete frequency peaks. The broadband noise is the reflection of pressure fluctuations caused by flow turbulence, viscous force, separation of the fluid boundary layer and vortex generated in the clearances between the rotor of a centrifugal pump and the adjacent stationary part of the casing. Meanwhile, mechanical sources similarly have contributions from the rotation of the pump shaft and bearings [24,25].

Turbulent perturbation seriously depends on the pump’s flow status. When a centrifugal pump operates under its design operation point, the maximum proportion of energy is used for the transfer of process fluid, the turbulent perturbation intensity would become the weakest. However, if the centrifugal pump operates under the smaller flowrate than its design flowrate, additional hydraulic noise would be generated due to the internal recirculation in both suction and discharge regions of the impeller. By contrast, if the centrifugal pump operates under the larger flowrate than that of its design, additional hydraulic noise would also be created as the increasing fluid boundary layer separation and the following periodic vortex shedding. Meanwhile, the discrete frequency characteristics occurring in the overall spectrum are basically attributed to the rotor blades, forming their discrete nature when interacting with their nearby stationary objects like volute tongue and periodic flow. These discrete components usually have frequency characteristics at the rotating frequency, the blade-passing frequency (BPF) and their integer frequency.

Considering that the turbulent perturbation intensity would reach the weakest level around the pump’s best performance point, the discrete frequency peaks incline to dominate the vibration spectrum, especially the low-order harmonics. If BEP during the operation of a centrifugal pump is not considered, the broadband noise results from turbulent disturbance might even exceed the tonal noise [26,27,28].

### 2.2. Development of the Single-Value Indicators of Vibration for a Centrifugal Pump

Dynamic characteristics of centrifugal pump could be reflected through statistical analysis. Therefore, relevant indicators could be established by means of statistical parameters.

The probability density of the data samples is defined as follows:(1)Prob[x≤x(t)≤x+dx]=p(x)dx

The expectation of a random function of time *x*(*t*) could be expressed as shown below:(2)$$E\left[x\right]=\int_{-\infty }^{+\infty }xp\left(x\right)dx,\;\overset{\infty }{\underset{i=1}{\sum }}x_{i}p_{i}$$

The rth-order moment about the mean $\overset{¯}{x}$ could be obtained through the following equation:(3)$$E\left[x\right]=\int_{-\infty }^{+\infty }{\left(x-\overset{¯}{x}\right)}^{r}p\left(x\right)dx$$

From Equation (3), it could be observed that the expectation of the random variable happens to be the first-order moment, the *RMS* value happens to be the square root of the second-order moment, and the variance is the second-order central moment. The *RMS* reflects the intensity of the variable, and the variance gives the deviation from the mean.

If the data is obtained in discrete form, then Equation (3) could be rewritten as:(4)$$M_{r}=\frac{1}{N}\sum_{K=1}^{N}{\left(x_{k}-\overset{¯}{x}\right)}^{r}$$

In this equation, N is the number of data samples, r represents the order of the moment.

The equations shown as follows demonstrate the calculation of *RMS* and standard deviation of variables in both continuous and discrete form:(5)$$RMS=\sqrt{\int_{-\infty }^{+\infty }x^{2}p\left(x\right)dx}=\sqrt{\frac{1}{n}\sum_{k=1}^{n}x_{k}^{2}}$$

Standard deviation:(6)$$\sigma =\sqrt{\int_{-\infty }^{+\infty }{\left(x-\overset{¯}{x}\right)}^{2}p\left(x\right)dx}=\sqrt{\frac{1}{n}\sum_{K=1}^{n}{\left(x_{k}-\overset{¯}{x}\right)}^{2}}$$

Complicated signal that might consist of a few sinusoids whose relative amplitudes and phases are unpredictable often need further processing. Under this circumstance, additional descriptions of signal like Crest factor (*C*_f_) and Kurtosis (*K*) could be very useful. Crest factor could be expressed as shown in Equation (7), where *V*_p_ represents the peak value of signal and *V_RMS_* represents its *RMS* value. Crest factor reflects the sharpness of the peaks occur in the signal, which could be employed to find whether the signal contains repeated impulses.
(7)$$C_{f}=\frac{V_{p}}{V_{RMS}}$$

Kurtosis is defined in Equation (8); −3 is included in the equation for normalization so that the kurtosis of Gaussian distribution is 0, which means the natural kurtosis of a Gaussian distribution before normalization is 3. Kurtosis would excess 0 if the peaks distributed in the spectrum are sharper than the Gaussian distribution. A higher value of kurtosis often indicates larger deviation. A negative value of kurtosis usually means the distribution presents much flatter than the normal. On the contrary, a positive value of kurtosis means the distribution is sharper than the normal.
(8)$$K=\frac{M_{4}}{\sigma_{4}}-3$$

Spectral entropy is described as a generalized form of information entropy by Pan et al. [29], which is usually used as a measurement for information uncertainty. Spectral entropy could be calculated through the following equation:(9)$$SE=-\sum_{i=1}^{N}p_{i}\cdot \log_{2}{(p}_{i})$$

In order to avoid the effects of data length, spectral entropy needs to be normalized for comparison [30]. More concretely, spectral entropy *SE* is normalized to *SE*_n_ according to Equation (10), where N represents the number of the data samples.
(10)$${SE}_{n}=-\frac{\sum_{i=1}^{N}p_{i}\cdot \log_{2}{(p}_{i})}{\log_{2}\left(N\right)}$$

Therefore, it could be found that the normalized spectral entropy *SE*_n_ has the value range from 0 to 1. The value of *SE*_n_ is larger when amplitude distribution is relatively flat. This is especially the case when each frequency component of the signal has the basically same amplitude, and the value of *SE*_n_ would reach the highest under that extreme condition. Conversely, *SE*_n_ would become lower when a few frequency components have the relatively large amplitude, and would have its minimum value of 0 when there only exists one non-zero amplitude frequency peak in the spectrum. From the nature of *SE*_n_ described above, it could be concluded that *SE*_n_ could be highly suitable for analyzing the spectral structure of centrifugal pump’s vibration signal.

## 3. Experiment

### 3.1. Test Bed

To test theoretical predictions against experimental results, an experimental rig was built for operation simulation of a centrifugal pump. As shown in Figure 1, the experimental rig was made up of a centrifugal pump, two water tanks, a suction line and a discharge line, which was constructed and formed as a close loop for water circulation.

The left tank shown in the figures, which was connected to the vacuum pump, was the cavitation tank used for cavitation simulation. Meanwhile, the right tank was a stable flow tank, which was employed to stabilize the flow. The test centrifugal pump was deployed in a horizontal plane, its outlet flange was 2 m below the water surface in the cavitation tank. Two stainless steel pipes were matched and connected to the suction and discharge flanges, which have the diameters of 60 and 50 mm and used to carry the upstream and downstream of the test pump respectively. Water temperature was maintained around 1 °C for at least one hour before each experiment. The design parameters of the centrifugal pump data are introduced in Table 1, and its performance curves are shown in Figure 2.

The operation condition of centrifugal pump could be considered as relatively stable when d*H*/d*Q* is negative, which means the pump’s head shows a downward trend as the flowrate becomes larger. Meanwhile, a positive value of d*H*/d*Q* means the centrifugal pump operates under a rather unstable condition. This unstable operation region indicates the performance of strong flow hydraulics result from the pump’s unreasonable design parameters. A centrifugal pump operating in such a region would strengthen the vibration intensity. As shown in Figure 2, d*H*/d*Q* at the range from 3 m^3^/h to 20 m^3^/h is positive, which indicates hydraulic instability of the pump.

Net positive suction head (*NPSH*) is the difference between the suction head and the liquids vapor head; the net positive suction head available from the application to the suction side of pump is often named *NPSH*a. *NPSH*r, called as the Net Suction Head as required by the pump in order to prevent cavitation for safe and reliable operation of pump. As suggested in ISO 3555, the predicted cavitation characteristics between the *NPSH* available (*NPSH*a) and the NPSH required (*NPSH*r) for the centrifugal pump could be achieved through throttling the valve in discharge line. Meanwhile, the rotational speed of the pump should be maintained at 2900 rpm, and the valve in the suction line should be fully opened.

The cavitation characteristic curve of the centrifugal pump obtained is shown in Figure 3. It could be clearly seen that cavitation occurs when the centrifugal pump operates around 64 m^3^/h. Moreover, full cavitation would be caused under approximately 82 m^3^/h while the pump head is approximately 8.8 m and the value of *NPSH*r is about to become larger than that of *NPSH*a. According to ISO 3555, when the pump’s flowrate keeps increasing and exceeds 82 m^3^/h, full cavitation would be caused. Meanwhile, as observed in Figure 3, cavitation performance of the centrifugal pump begins to deteriorate rapidly under the flowrate of 65 m^3^/h onwards. Hence, the range of pump’s flowrate from about 65 m^3^/h to 82 m^3^/h could be regarded as the cavitation development phase, in which the cavitation would become continuously worse as the flowrate increases.

### 3.2. Data Acquisition System

A data acquisition system was built to acquire the centrifugal pump’s performance parameters and vibration signal in real time. Sensors used to monitor the pump’s performance parameters mainly included the DM4022 induction tachometer (YINHE ELECTRIC, Changsha, China) for rotational speed measurement, LDG-SIN-CN65-Z2 electromagnetic flowmeter (Asmik, Hangzhou, China) which was installed in the discharge line, WIKA S-10 pressure sensors (WIKA, Bavaria, Germany) for pump head calculation which were installed in the suction line and discharge line, respectively, and PCB MA352A60 vibration sensor (PCB, New York, NY, USA) for vibration signals measurement. Their main parameters are shown in Table 2 and their installation scenes are shown in Figure 4.

Moreover, a PXI-4472 dynamic signal acquisition card (NI, Austin, TX, USA) was employed for data calculation and information aggregation. Its design parameters are introduced in Table 3. Its software composition is shown in Figure 5.

### 3.3. Experimental Procedures

Vibration of the centrifugal pump was evaluated under the following two circumstances:

1. A centrifugal pump operating under different flowrates but a fixed rotational speed of 2900 rpm. The pump’s flowrate was changed by progressively adjusting the opening degree of the throttling valve in the discharge line.

2. A centrifugal pump operating under different *NPSH*_A_ but at a fixed flowrate and a fixed rotational speed. The inlet pressure was adjusted by the vacuum pump.

In order to obtain more reliable experimental results, experiments were repeated three times, sufficient time interval existed between each test. The obtained data was processed and analyzed by Matlab.

The sample time was selected at 500 rotation cycles of the test pump. Samples of the vibration signal obtained under three typical flowrates are shown in Figure 6.

Figure 7 shows the vibration frequency spectra under five flow conditions, which represents five typical operation conditions of the centrifugal pump, namely, part load operation condition, efficient operation condition, design operation condition, progressive cavitation, and fully cavitation.

Therefore, the data sets are explored in the frequency domain. Figure 7 shows the spectra for the five types of measurement under typical flow rates that correspond to partload operation, efficient operation, design operation, progressive cavitation and full cavitation. It can be seen in the figure that vibration shows a clear change in spectra with different flow rate acoustics. In particular, vibration spectra show that there is a steady increase in amplitude above the frequency of 3 kHz whereas amplitude in the low frequency range is in a reverse. This is consistent with the vibration generation processes because high cavitation will produce more random noise. In the meantime, the discrete components will be reduced due to the random fluid oscillations.

## 4. The Analysis of the Single Value Operation Monitoring Indicators for Centrifugal Pump

### 4.1. Single-Value Indicators of Vibration from Time Domain

In terms of experimental analysis, accelerometers mounted on the centrifugal pump three direction as shown in Figure 4, In order to simplify, the X direction data is mainly used in the indicator research

Figure 8 and Figure 9 show the peak and *RMS* trend of the time domain vibration signal with the flowrate from 0 to 1.6 *Q*_d_.

Both peak and *RMS* could reveal the change law of signal energy intensity, these two indicators share a similar tendency with the pump’s flowrate as shown in the figures above. Both indicators would rapidly increase when pump’s flowrate exceeds 60 m^3^/h, which may also be the inception cavitation point according to Figure 3.

However, it was found that the change tendency of the *RMS* was much more flatter than that of the peak, especially during the flowrate range from 10 m^3^/h to 18 m^3^/h, which is the hump area of the pump performance curve. In this area, a relatively strong flow instability phenomenon exist in the pump, peak curve has a relatively steep rise trend, yet this trend is not observed in the *RMS* curve. Meanwhile, the peak value would become extreme low at 53 m^3^/h, which indicates the highest efficiency operation point of the pump, and such tendency is also not shown in *RMS* curve. Furthermore, the start point of the dramatic rise of the peak occurs earlier than *RMS*. From these analyses, it might be concluded that the peak value is more sensitive in flow instability detection than *RMS*. By contrast, *RMS* performs better in predicting cavitation, it has strong anti-jamming capability contrast to low sensitivity.

Crest factor is described as the peak factor divided by the corresponding *RMS* value. As the high detection sensitivity of peak and the strong anti-jamming capability of *RMS*, the crest factor might be highly suitable for flow instability detection. As shown in Figure 10, an interesting phenomenon could be found whereby the extreme points of the crest factor curve are the pump’s particular operating points, which show the flow instability boundary, cavitation inception and even the highest efficiency point.

Figure 11 shows the kurtosis curve for the time domain vibration signal, and two different definite increase could be easily found in the kurtosis curve. The first was observed at the operation range of 10–18 m^3^/h, the other was between 55–62 m^3^/h. The first increase indicates the hump area of the pump performance curve; in this area, the internal flow status of the centrifugal pump is rather unstable. The second increase locates at the onset of the cavitation, which also means stronger intensity of flow instability.

As can be seen from Figure 7, the probability density function (*PDF*) of the vibration signal is approximately Gaussian with the exact shape depends on its details. As the flowrate increases, the vibration would change and, obviously, the shape of the *PDF* curve would also change correspondingly.

The shape of the *PDF* curve could be evaluated by standard deviation and variances, and as a result, both mathematical parameters could be used as the single value indicators.

The change tendency of the standard deviation and the variances for the time domain vibration signal with the pump’s flowrate are shown in Figure 12 and Figure 13. It could be found that the standard deviation curve and the variances curve have almost the same variation law with the *RMS* curve, thus add nothing qualitatively new to the discussion.

Figure 14 shows the amplitude of the *PDF* curve for the time domain vibration signal under different flowrates, which clearly shows that there is a broad maximum value of the *PDF* exists at the flowrate range from 50 m^3^/h to 55 m^3^/h. When the pump’s flowrate exceeds this range, a definite decline trend occurs in the curve. Such a change law is confirmed as a universal phenomenon, and this turning point appears in the curve could detect the likely beginning of cavitation. However, the flow instability which should be reflected in the range of 10–17 m^3^/h is not shown, obviously.

The reason for such a pattern of the *PDF* is that the distribution of the vibration signal in terms of frequency would change with the pump’s operation condition. As the flowrate keeps increasing and the pump approaches cavitation, it could be seen that variations occur in the spectral structure of the vibration signal, where under the relatively low flowrates the vibration spectrum includes a few isolated low-level structural resonances. When centrifugal pump operates under the established cavitation detection point, a number of high peaks appear in the spectrum, where the *PDF* for the time domain vibration signal has its sharpest peak. When the centrifugal pump operates under the highest flowrate during the test, which means a severe cavitation condition, a large number of peaks appear across the spectrum, where the *PDF* for vibration is inclined to flatten out. Meanwhile, it should be noted that some other flow instability phenomena like backflow would cause stronger broadband noise in some areas, which would result in the change of the *PDF* pattern for the vibration signal. However, the degree of deformation of the *PDF* curve is weaker than the cavitation caused. Therefore, the *PDF* curve could be employed for cavitation diagnosis for centrifugal pump.

### 4.2. Single-Value Indicators of Vibration from Frequency Domain

Previous investigations suggest that the vibration spectral of a centrifugal pump contains a broadband noise and a few discrete frequency peaks, whose intensity would be strongly affected by the flow status of the centrifugal pump. In the analysis of the frequency domain vibration signal, crest factor, kurtosis and entropy might possess the ability to evaluate the intensity change for the two components in the frequency domain.

As shown in Figure 15 and Figure 16, the change regularity of these two statistical parameters with the pump’s flowrate is similar, both parameters could clearly reflect the hump area and the cavitation onset by the curve drops, and indicates the cavitation process with a relatively low value.

Figure 17 shows the spectral entropy obtained from the frequency domain vibration signal as a function of the pump’s flowrate. The value of the spectral entropy for the frequency domain vibration signal would be expected to be relatively low in the area centered at the pump’s design flowrate, where the harmonic components of *BPF* and rotational frequency stand out above the spectrum, and information uncertainty is lowest. As the pump’s flowrate deviates from its design, spectral entropy would increase correspondingly due to the stronger noise. The vibration signal would increasingly resemble broadband noise and spectral entropy would finally have its highest value. The curve obtained confirms this general analysis, and further indicates that the spectral entropy of the frequency domain vibration signal could detect cavitation onset and unstable flow area of centrifugal pump.

### 4.3. Comparison of Single-Value Indicators in Multiple Measurement Directions

As mentioned in a previous article, the use of a single sensor is very important for the promotion of monitoring technology, Therefore, in the research and analysis of the establishment of the previous indicator, only the X-based direction was considered because the much more easy installation and the transmission of the signal.

As shown in Figure 18 and Figure 19, the trends of several directions in the time domain and frequency domain with the operating conditions are almost the same, but the indicator values are slightly different. It can be considered that the vibration index detection is applicable to different directions, so in practical applications only the convenience of installation and signal transmission need to be considered

## 5. Conclusions

This paper explores the characteristics for the vibration signal of a centrifugal pump through statistical analysis, and some conclusions that are drawn are as follows:

1. Dynamic characteristics of the centrifugal pump’s performance characteristics would change under unstable flow status, which would be reflected in the vibration signal, and as a result statistical analysis for vibration signal could be employed as an effective approach to detecting off-design operation conditions for a centrifugal pump.

2. From the statistical analysis for the time domain features of vibration signals, it is concluded that the rapid increase of the peak and *RMS* value could indicate the unstable flow status of a centrifugal pump. The peak indicator demonstrates a higher capability in detection. Most extreme points in the crest and kurtosis curve have definite physical significance, which might be the onset and the end of the flow instability area, the onset of cavitation or the highest efficiency operation point. Meanwhile, the shape or pattern of the probability density function (*PDF*) for the pump’s vibration signal performs well in the detection of the early stage cavitation.

3. From the statistical analysis for the frequency domain features of the vibration signal, it is found that the kurtosis, crest factor, and entropy are very suitable for detecting the intensity change of the broadband noise and discrete frequency components in the frequency domain which are caused by the variation of the pump’s flow status.

## Figures and Tables

**Figure 1 sensors-20-03283-f001:**
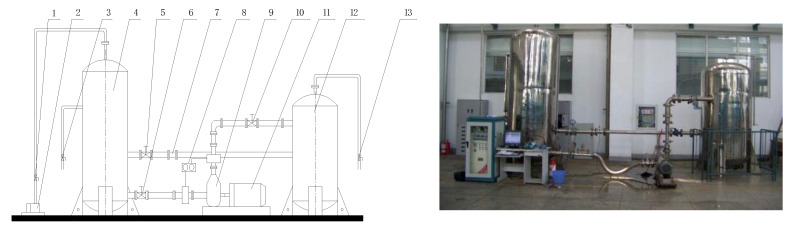
Schematic of the experimental rig for operation simulation of centrifugal pump. 1, 3, 13. Ball valve; 2. Vacuum pump; 4. Cavitation tank; 5, 10. Butterfly valve; 6. Gate valve; 7. Electromagnetic flowmeter; 8. Pressure transmitter; 9. Model pump; 11. Motor; 12. Stable flow tank.

**Figure 2 sensors-20-03283-f002:**
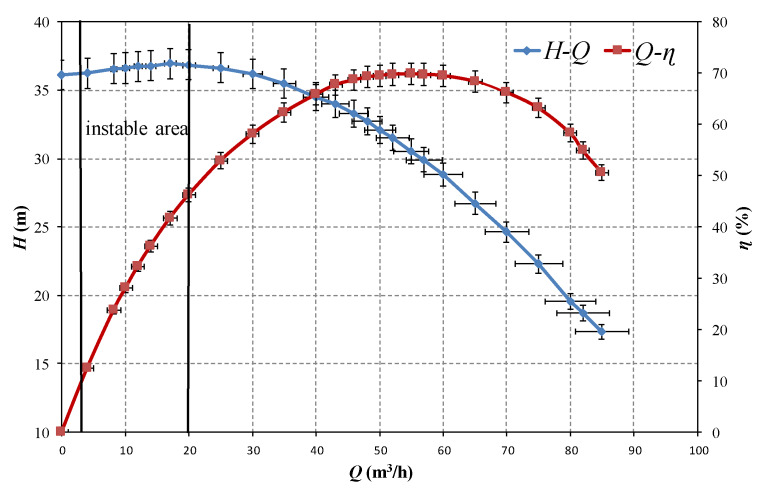
Pump performance curve.

**Figure 3 sensors-20-03283-f003:**
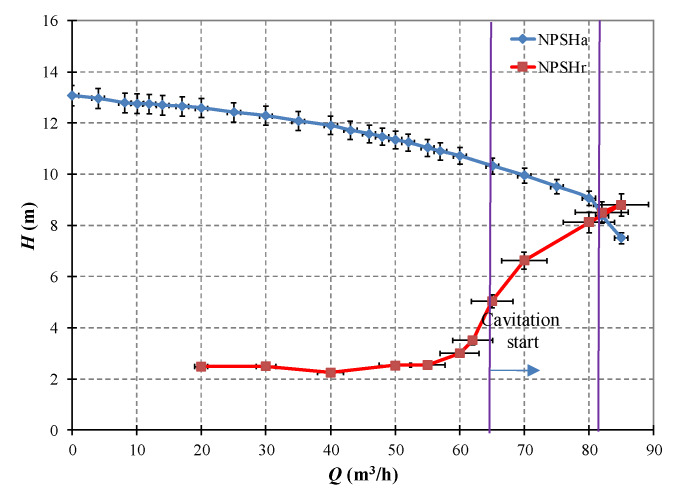
Cavitation characteristic curve of the centrifugal pump.

**Figure 4 sensors-20-03283-f004:**
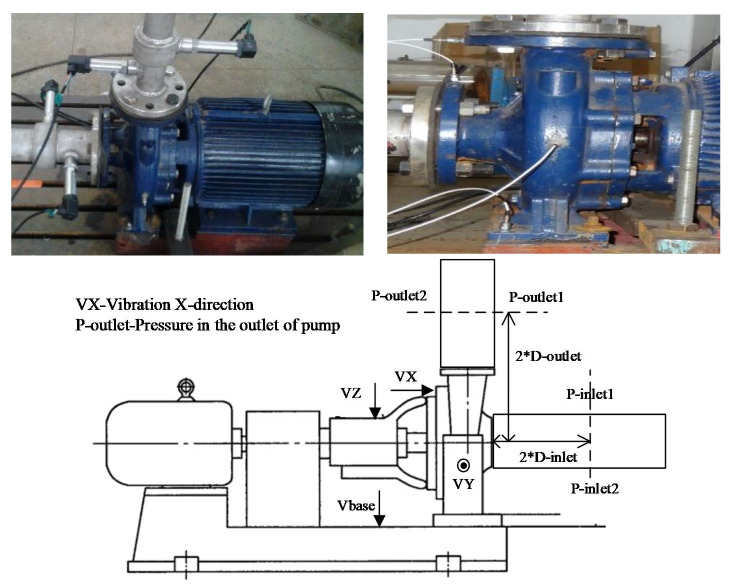
Performance monitoring sensors installation.

**Figure 5 sensors-20-03283-f005:**
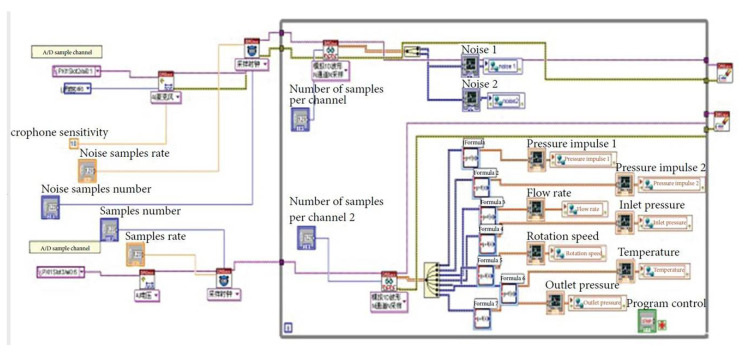
Software composition of the dynamic signal acquisition card.

**Figure 6 sensors-20-03283-f006:**
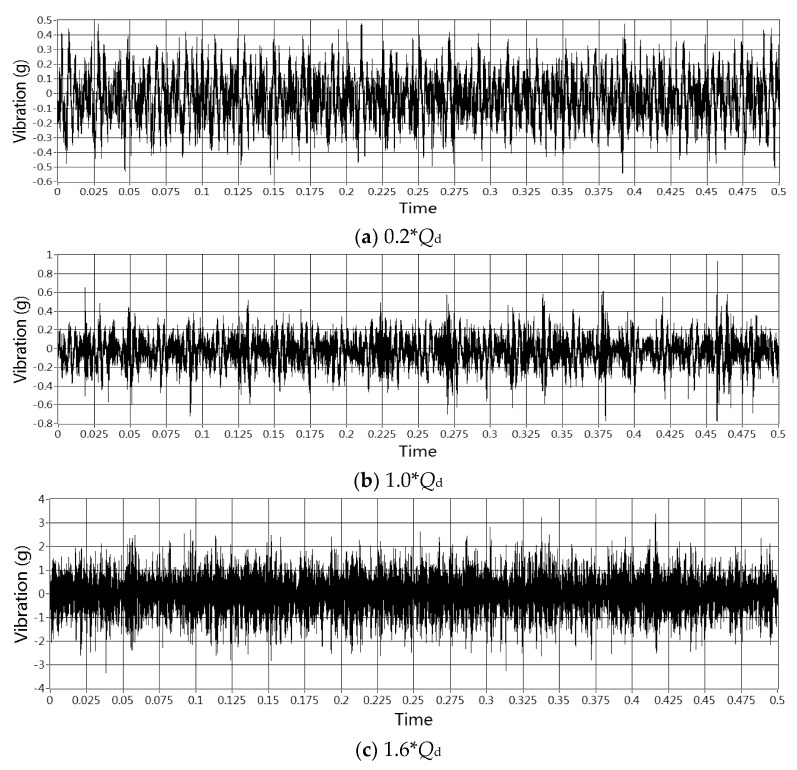
Raw time domain vibration signal under three typical flow conditions.

**Figure 7 sensors-20-03283-f007:**
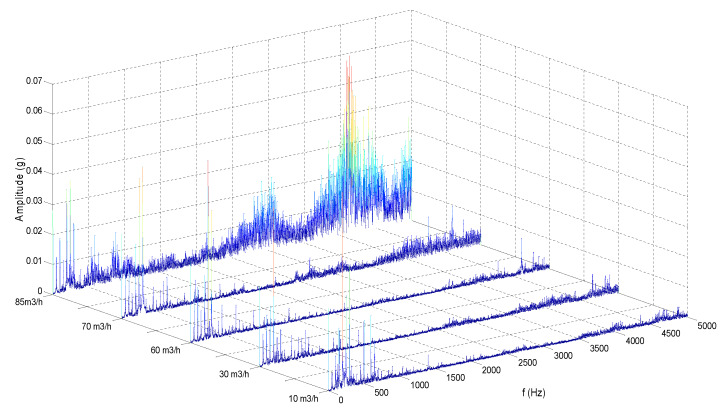
Vibration spectra of the centrifugal pump under five typical flow conditions.

**Figure 8 sensors-20-03283-f008:**
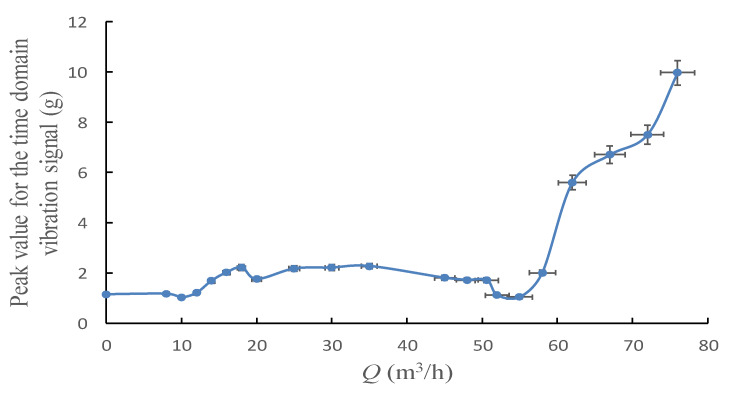
Peak for the time domain vibration signal under different flowrates.

**Figure 9 sensors-20-03283-f009:**
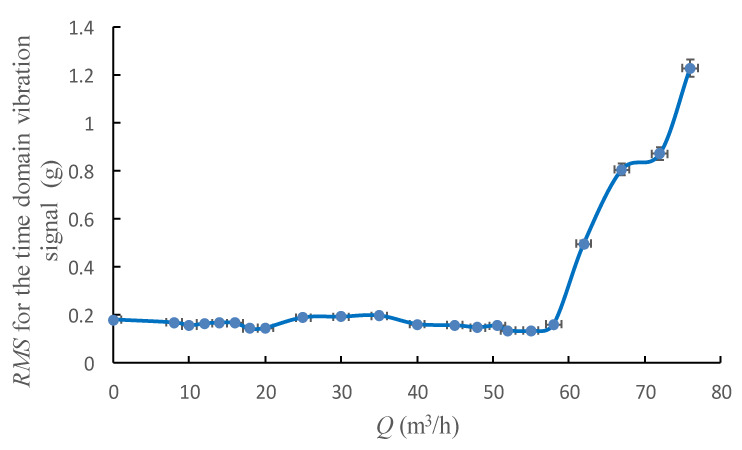
Root mean square (*RMS)* for the time domain vibration signal under different flowrates.

**Figure 10 sensors-20-03283-f010:**
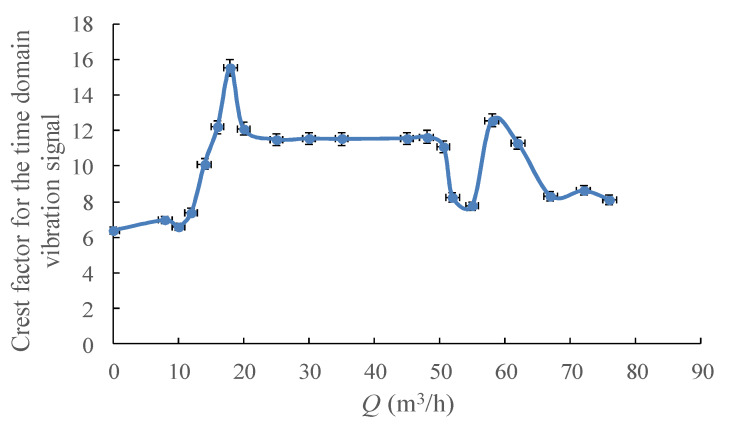
Crest factor for the time domain vibration signal under different flowrates.

**Figure 11 sensors-20-03283-f011:**
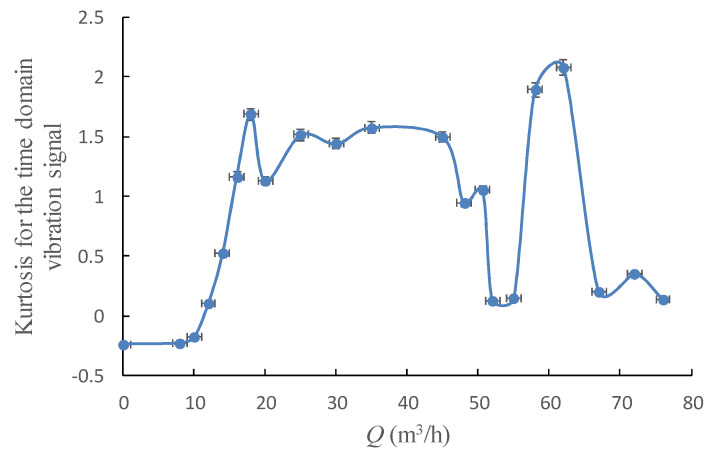
Kurtosis for the time domain vibration signal under different flowrates.

**Figure 12 sensors-20-03283-f012:**
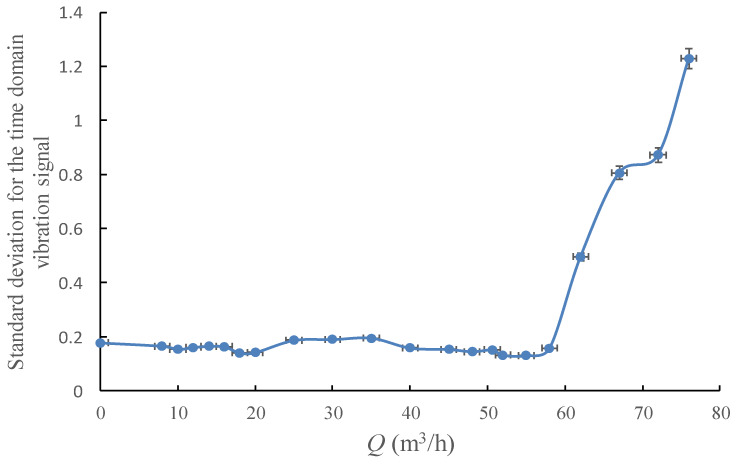
Standard deviation for the time domain vibration signal under different flowrates.

**Figure 13 sensors-20-03283-f013:**
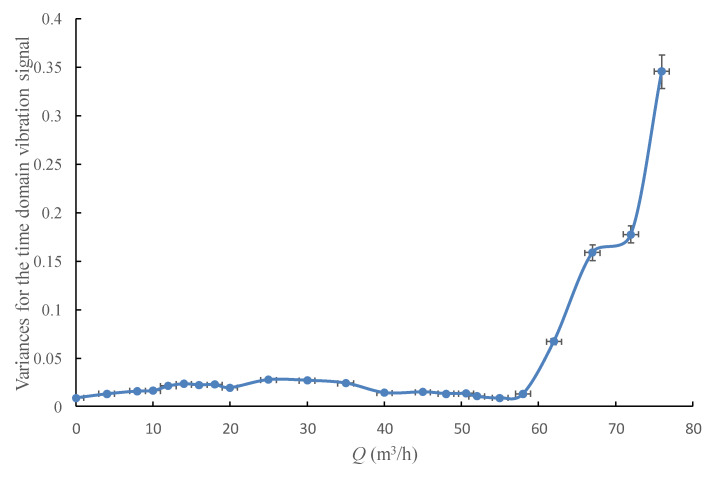
Variances for the time domain vibration signal under different flowrates.

**Figure 14 sensors-20-03283-f014:**
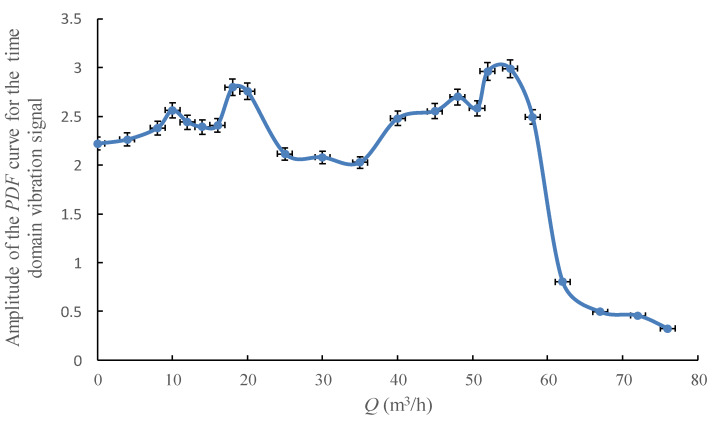
Amplitude of the probability density function (*PDF)* curve for the time domain vibration signal under different flowrates.

**Figure 15 sensors-20-03283-f015:**
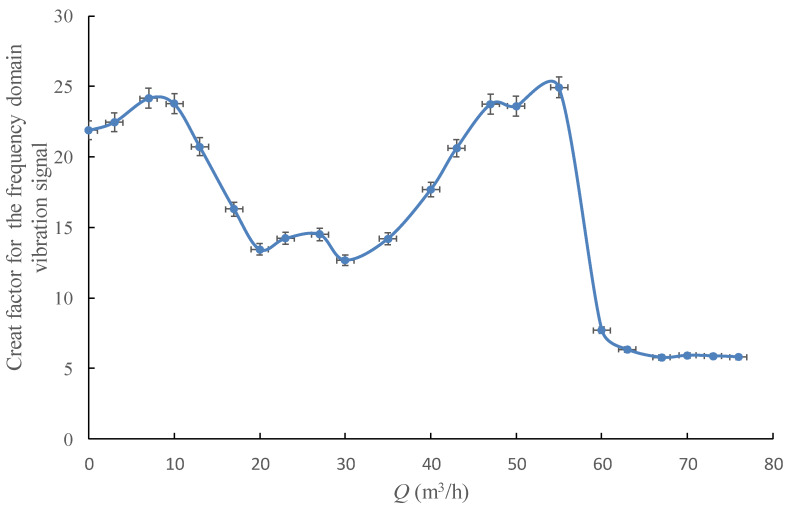
Crest factor for the frequency domain vibration signal under different flowrates.

**Figure 16 sensors-20-03283-f016:**
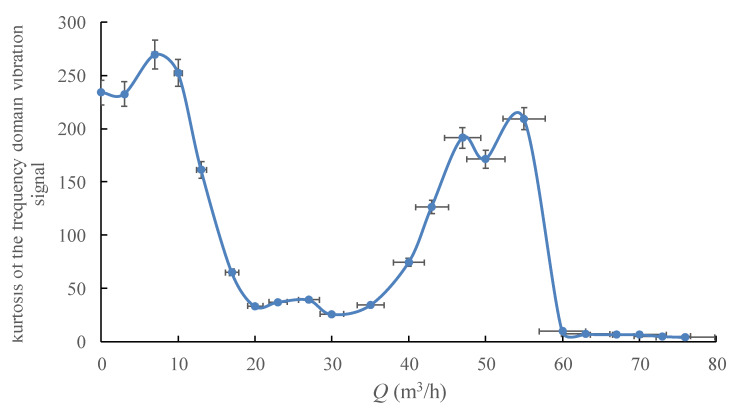
Kurtosis for the frequency domain vibration signal under different flowrates.

**Figure 17 sensors-20-03283-f017:**
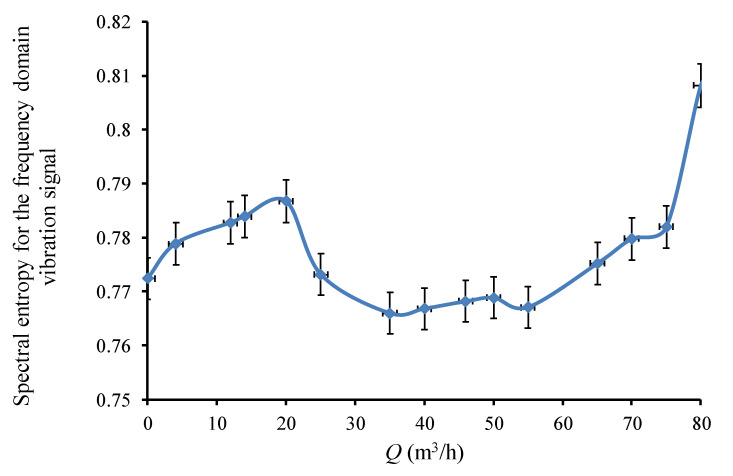
Spectral entropy for the frequency domain vibration signal under different flowrates.

**Figure 18 sensors-20-03283-f018:**
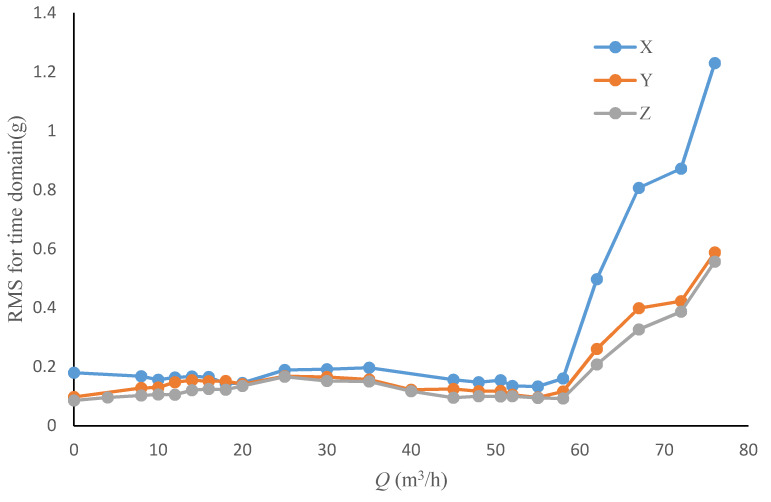
*RMS* for the different direction time domain vibration signal under different flowrates.

**Figure 19 sensors-20-03283-f019:**
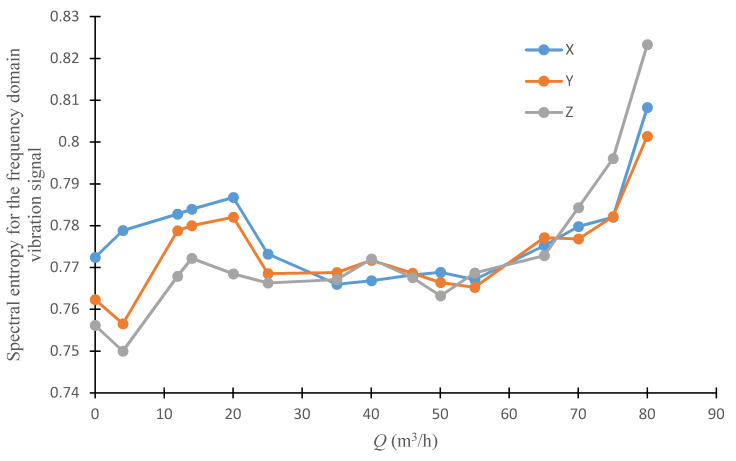
Spectral entropy for the different direction frequency domain vibration signal under different flowrates.

**Table 1 sensors-20-03283-t001:** Main parameters of the test centrifugal pump.

Parameter	Value
Rated flow (*Q*_d_)	50 m^3^/h
Rated head (*H*_d_)	32 m
Rated speed (*n*)	2900 r/min
Efficiency (*η*)	72%
Specific speed (*n*_s_)	101
Pump inlet diameter (*D*_s_)	60 mm
Impeller inlet diameter (*D*_j_)	75 mm
Impeller outlet diameter (*D*_2_)	174 mm
Blade width (*b*_2_)	12 mm
Blade number (*Z*)	6
Volute base diameter (*D*_3_)	184 mm
Pump outlet diameter (*D*_d_)	50 mm

**Table 2 sensors-20-03283-t002:** Main parameters of the sensors.

Sensor	Parameter	Value
DM4022 induction tachometer	Measurement range	0–3000 rpm
	Measurement accuracy	0.1%
LDG-SIN-CN65-Z2 electromagnetic flowmeter	Measurement range	0–100 m^3^/h
	Measurement accuracy	0.5%
WIKA S-10 pressure sensors	Measurement range	(inlet) 0–1.6 (bar)/(outlet) (1–4 bar)
	Measurement accuracy	0.2%
PCB MA352A60 vibration sensor	Measurement range	5 Hz–60 kHz
	Measurement accuracy	10 mv/g

**Table 3 sensors-20-03283-t003:** Design parameters of the data acquisition card.

PXI-4472—Dynamic Signal Acquisition	PXI-6251—High-Speed M Series Multifunction DAQ
Resolution	24-bit	Resolution	16-bit
Dynamic range	110 dB	Analog inputs	32 channel (1.25 MS/s)
Sampling rate	102.4 kS/s maximum	Analog outputs	2 channel (2.5 Ms/s)
Range	±10 V	Range	±100 mV to ±10 V
IEPE conditioning	Software configurable	Synchronization	Multiple-device
Synchronization	Multiple-device

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
