# Peer review of "Research on the Single-Value Indicators for Centrifugal Pump Based on Vibration Signals"

_sensors, 2020, doi:10.3390/s20113283_

Round 1

Reviewer 1 Report

This article analyses indicators of vibration for a centrifugal pump. The results obtained seem valuable but some corrections and clarifications must be made before I can recommend its publication in this journal:

The title is a bit misleading and has a weird phrasing.

Line 13: ‘more incline’: weird phrasing

Line 73: the authors claim ‘developing dimensionless parameters’, but as far as I am concerned, they use existing and already developed statistical parameters to analyse the problem. I am not implying that the study itself is not useful but the authors should tone down this statement.

Line 76: ‘is consist’: weird phrasing

Figure 2: what is the curved part of the piping next to the pump for?

Line 180: it is said that ‘dH/dQ at the range from5m3/h to 20m3/his positive, which indicates hydraulic instability of the pump, but in figure 3 the vertical solid lines were ‘instable area’ is read seem to indicate 3 to 8 m3/h.

Lines 184-185: Net Positive Suction Head Available (NPSHA) and the Net Positive Suction Head Required (NPSHR) meaning should be explained and defined. For instance, Figure 4 shows a NPSHA of 12m at 40 m3/h but the head in Figure 3 at 40 m3/h is 35 m. What is the difference due to?

Line 189: it is said that ‘cavitation occurs when the centrifugal pump operates around 82m3/h’ but Figure 4 shows that the cavitation starts around 64 m3/h and this is not mentioned in the text.

Lines 197-205 a table detailing the sensors used is encouraged (pressure sensors, accelerometers, etc)

Reviewer 2 Report

Dear authors, please find my recommendations regarding the paper "Established of the Single-value Indicators of Vibration for Centrifugal Pump"

  • first off all please revise the English, a few mistakes can be found in lines: 64, 153-154, 155-156 and others
  • please explain why the experimental tests have been performed at 1 degree Celsius,
  • for Fig. 3, please add the error bars,
  • on Fig. 3, please correct "instable area", the range of the unstable area is not linked correctly according with the text,
  • the text from Fig. 8, cannot be seen, please increase the text,
  • for Fig. 9, add the same scale for Y axis - Vibration,
  • for Fig. 10, please increase the text and the scales of the graph. A discussion related to this graph should be added. Which is the relevance of the low frequencies in terms of vibrations and which is the relevance of higher frequency for the maximum flow rate for example.
  • for Fig. 11-16, the Y axis has units ? Is possible to add some errors in the graphs, taking into account that the results are obtained experimentally.
  • in terms of experimental analysis, the accelerometers mounted on the centrifugal pump, how many directions can measure and which is the direction of measurement for your paper. If the experimental analysis was performed in only one direction, what is happening in the rest of two directions. Please add a comment regarding this remark. 

Reviewer 3 Report

The paper entitled “Established of the Single-value Indicators of Vibration for Centrifugal Pump” under review deals with the research on the possibility of applying the single-value indicators of vibration into operation condition monitoring for centrifugal pump. I see two elements of the work: 1) monitoring and 2) predication on off-design operation conditions for the pump. The article tackles an important issue in energy, fluid flow, pipes, reactors etc. The experimental part of the article contains crucial information regarding the research model, method and experimental technique. It also includes the information on the available equipment and presents the research outcomes and their detailed description. A structure of the paper is in accordance with principles of very good scientific reports. The paper is written in good English. The article contains adequate and appropriately selected 17 literature items and 20 figures.

Comments:

Fig. 1 shows experimental setup – in my opinion photo of the setup (Fig. 2) is unnecessary,

Table 2, main text – units by SI should be used,

Figs 5 and 6 are unnecessary, it would be better to present a schematic of the arrangement of the sensors,

in my opinion Fig. 7 is unnecessary,

Figs 11-20 – should be point plots (no lines), additionally the authors should give the error bars. Statistics are the weak point of this work.

In opinion of the reviewer the article can be accepted for publication after minor revision.
